# OpenReview forum: "Watch Out for Your Agents! Investigating Backdoor Threats to LLM-Based Agents"
_NeurIPS.cc/2024/Conference — NeurIPS 2024 poster_

### Official Review · Reviewer_9U6V · 2024-07-01

**Soundness:** 3
**Presentation:** 2
**Contribution:** 3
**Rating:** 7
**Confidence:** 3

**Summary:**

This paper focuses on backdoor attacks on LLM-based agents via poisoning of the fine-tuning data. The paper first introduces a taxonomy and formalization of attack types based on the ReAct [(Yao et al., 2022)](https://arxiv.org/abs/2210.03629) paradigm. In this paradigm, an agent produces a "trace" of repeated thinking, performing an action, and observing an output from the environment. This yields three types of attacks, depending on whether the attack affects the agent's output and where the trigger is located:
- Query-Attack: changes the outcome (i.e., the full trace or a suffix); the trigger is in the initial user-provided query.
- Observation-Attack: changes the outcome (i.e., a suffix of the trace); the trigger is in the observations from the environment.
- Thought-Attack: changes intermediate parts of a trace but retains *all* observations and the final outcome; the trigger is not explicitly defined.

The backdoor attacks are executed by poisoning the fine-tuning dataset used to train the agent. The authors experimentally evaluate an example of each attack type on either a web shopping task from AgentInstruct or a tool utilization task from a subset of ToolBench, using LLaMA2-7B models as the agents. Results show that all attack types are effective.

Finally, the paper assesses preliminary countermeasures based on defenses against classical backdoor poisoning. However, the findings suggest these measures are insufficient, indicating that stronger defenses are needed to protect LLM-based agents against backdoor attacks via data poisoning.

**Strengths:**

The authors provide a holistic picture of backdoor attacks for LLM agents. Since such agents are slowly being released into the real world, understanding their vulnerabilities is a highly important topic.

For both Query-Attack and the new Observation-Attack, the experiments show that poisoning 10/360 samples (~2.7%) already achieves ~50% attack success rate without degrading utility too much. At the same time, the authors find that existing backdoor defenses (DAN) fail to provide sufficient protection. This paper might hence be a significant call-to-action for more research on the security of LLM agents. The new taxonomy of attack types is a helpful tool for this.

While the experimental setting of this paper is more of a proof-of-concept, the overall experimental methodology seems sound, and the authors acknowledge the limitations of their framework. Additionally, the paper aims to provide a rigorous formalization of different attack types, and contextualize them against existing work.

**Weaknesses:**

Edit: After clarifications and additional results from the authors, all attacks (in particular, Thought-Attack) look stronger than I initially thought. I increased my score accordingly.

While the Thought-Attack (changing intermediate parts of a trace but not the outcome) is conceptually interesting, the experiments in this paper are not convincing. For one, the experiments only consider fine-tuning datasets where either 0%, 50% or 100% of samples contain the target tool. Poisoning ratios of >50% are not realistic. Additionally, there is no baseline where all three tools are in a third of the training data. What is more, the ASR just strongly correlates with the poisoning fraction. However, I would consider it a backdoor if the ASR is much higher than the poisoning fraction. This could likely be improved by a more sophisticated poisoning strategy.

Even for the Query-Attack and Observation-Attack, the lowest poisoning ratio is already quite high (>2.7%). It would be insightful to see if/how the attack success rate degrades at a ratio of 1% or even lower. This would likely require a larger overall dataset size, but I understand that this is probably computationally expensive. Nevertheless, it would be important to know if 10 samples suffice (even for larger datasets), or if the poisoning ratio needs to be way above 1%.

Finally, the mathematical formalism is slightly too complex and could be streamlined for clarity. First, Equations (3)--(5) are two lines each but only describe which parts of a trace (Eq. (2)) is targeted. It also seems that there has been a LaTeX error in the second lines of Eqs. (3) and (4). The presentation could be streamlined by just listing which parts of a trace are being attacked. Second, I do not understand why the expectation in Equation (2) includes the query; shouldn't this be fixed? All the quantities in Eq. (2) are fixed, hence it is not clear why the expectation is necessary at all. Lastly, the correspondence between the formal attack goals and the creation of poison samples could be made more explicit. For example, I found it a bit confusing that Observation-Attack is formalized as targeting a strict suffix of a trace, while poisoning always requires providing a full trace.

Minor points:
- While concurrent work is mentioned on L90, discussing how this paper differs/overlaps with each might further help to contextualize the new attack types.
- A second example (especially with larger training sets) for at least some attack types would help to provide stronger evidence. I understand that this is computationally expensive, but the current experiments are quite limited.
- Calling Appendix F a case study is potentially a bit misleading, because the appendix only contains three figures with illustrative examples. Those figures are insufficient to provide conclusive evidence. Nevertheless, the illustrative examples are helpful.
- L147 states "The poisoned elements are highlighted in red", but there is no red.

**Questions:**

1. What exactly is the randomness for the expectations in Equations (2)--(5)?
2. In 3.2.2, after (1.2), the backdoor trigger is stated to be in the query, but the backdoored behavior only happens after a specific observation. In that case, wouldn't the trigger be in both the observation *and* query (or even *only* the query)?
3. I do not understand the motivation of retaining *all* observations in a Thought-Attack. Wouldn't it suffice to retain the final output?
4. Did the authors observe some negative results, i.e., cases in which poisoning failed (i.e., either degraded utility too much or failed to create a backdoor)?
5. Why are the two "Clean" rows in Table 1 different from Table 2? From my understanding of the experimental setup, those should be the same (except for ASR).

**Limitations:**

The authors very transparently discuss the limitations of their work, especially the limitation that they only consider one agent paradigm and only evaluate one dataset per attack type.

---

> ### Author Rebuttal · Authors · 2024-08-07
>
> We sincerely thank you for your careful reviewing. We make the following response to all your questions.
>
> **Q1:** Regarding the poisoning ratios used in Thought-Attack.
>
> **A1:** The detailed response to this question is in the global Author Rebuttal. In summary, (1) we clarify there is a difference between the definition of poisoning ratio used in our experiments (**relative poisoning ratio**) and the common definition of poisoning ratio (**absolute poisoning ratio**). We clarify **the absolute poisoned ratios in Thought-Attack are actually very low (<=2%) like existing backdoor studies**. (2) We explain why the relative poisoning ratio can achieve even 100% in Thought-Attack. (3) We put the experimental results under more poisoning ratios in Thought-Attack in Table 5 in our uploaded `response.pdf`.
>
> **Q2:** Regarding using smaller poisoned ratios for Query/Observation-Attack.
>
> **A2:** (1) First ,we provide the results of mixing agent data with general conversational data from ShareGPT in Table 1 in our uploaded `response.pdf`. **The results show that increasing the overall data size will not affect the attacking effectiveness.**
>
> (2) Then, we conduct experiments with only 5 poisoned examples for Query/Observation-Attack, which decreases the poisoning ratio to ~1.4%. The results are in Table 2 in our uploaded `response.pdf`. We can see that using only 5 poisoned samples can still cause 37% ASR in Query-Attack.
>
> **Q3:** Regarding the suggestion “the presentation could be streamlined by just listing which parts of a trace are being attacked.”
>
> **A3:** Thank you for your suggestion. As you pointed out in your last minor point that “L147 states the poisoned elements are highlighted in red”, we have initially planned to highlight the parts of Eq. (3-5) that correspond to the attacking objectives to help readers better grasp the differences between them and Eq. (2), but we have missed doing so. We will fix them in the revision.
>
> **Q4:** Regarding the randomness for the expectations in Eq. (2-5).
>
> **A4:** In Eq. (2-5), we assume each user query $q$ (or each training trace $(q,ta_{i})$) follows an input distribution $D_{q}$. Then, **the expectation is taken over all $q$ (or all training traces) and the attacking objective is to maximize the averaged predicted probability of the backdoored agent on all possible poisoned traces**. We will mention this in the revision.
>
> **Q5:** Regarding the question “... Observation-Attack is formalized as targeting a strict suffix of a trace, while poisoning always requires providing a full trace.”
>
> **A5:** The prefix trace before the backdoor is triggered in Observation-Attack is crucial because it ensures that the model learns the pattern that the backdoor is activated only after the trigger appears in a specific observation instead of at the beginning.
>
> **Q6:** Regarding the detailed discussion on the concurrent work.
>
> **A6:** The discussion in Section 3.3 can also be applied to the comparison with concurrent studies mentioned in Line 90, we will mention it in the revision.
>
> **Q7:** Regarding the case studies.
>
> **A7:** **The examples in Appendix F are real cases** as the texts in figures are exactly the original model responses and environment feedback on testing queries, rather than imaginary examples.
>
> **Q8:** Regarding the situation (1.2) in Section 3.2.2.
>
> **A8:** In the situation (1.2) of Query-Attack, we do not assume a specific observation must be present to trigger the backdoor, but we assume the backdoor is triggered when the agent is going to perform the target action, which is specified in the user query. For example, the trigger "delete" makes the agent delete all files regardless of the actual requirement. Thus, the trigger is only in the query.
>
> **Q9:** Regarding the question “... the motivation of retaining all observations in a Thought-Attack. Wouldn't it suffice to retain the final output?”
>
> **A9:** We agree with you that in real cases, Thought-Attack does not need to retain all observations. We made this assumption in the paper mainly for two reasons: (1) As Eq. (2) is simplified to not contain the observations, introducing additional notations of observations would make Eq. (5) more complex and harder to understand. (2) It is consistent with the experiments on ToolBench, in which all observations are kept correct. But we will revise the statement in Line 171 accordingly.
>
> **Q10:** Regarding the possible negative results.
>
> **A10:** One potential negative result is about the degradation of the Reward scores on WS Target, which is analyzed in Line 279-287.
>
> **Q11:** Regarding the question “Why are the two "Clean" rows in Table 1 different from Table 2?”
>
> **A11:** Thank you for your insightful question.
>
> (1) First, **the two models “Clean” in Table 1 and Table 2 are the same model**.  As you can see, the results on AW, M2W, KG, OS, DB and WS Clean are the same. **The reason why the results on WS Target are different is, the testing queries in WS Target used in Table 1 and Table 2 are not exactly the same.** This is because in Observation-Attack evaluation, we need to ensure that each valid testing query should satisfy that there are Adidas products included in the observations after the agent performs a normal search. Otherwise, the query will never support a successful attack. Therefore, we make a filtering for the testing queries used in Table 2.
>
> (2) Second, **the two models “Clean$^{\dagger}$” in Table 1 and Table 2  are not the same**. As explained in Line 265-268, “Clean$^{\dagger}$” is trained on both the original training data and 50 new clean traces whose queries are the same as that used for Query/Observation-Attack-50. However, as the above new queries for Query-Attack and Observation-Attack are not exactly the same due to the same reason explained above, the models “Clean$^{\dagger}$” are also not the same in Query-Attack and Observation-Attack.
>
> We will add the above clarification in the revision to avoid misunderstandings.

---

> > ### Comment · Reviewer_9U6V · 2024-08-08
> >
> > I thank the authors for their detailed response and especially for the additional experiments. The rebuttal answered all my questions. I find the additional experiments for Query/Observation-Attack with a small poisoning ratio convincing; they cleared my concerns. Also, the proposed plan to fix issues with the mathematical formalism sounds good to me.
> >
> > **Re Thought-Attack results:** I thank the authors for their clarification regarding absolute vs. relative poisoning ratios. I now agree that the absolute number of poison samples is reasonable.
> > But one concern remains: `... the ASR just strongly correlates with the poisoning fraction.` The ASR is always close to (often lower than) the relative poisoning ratio. I would expect that if x% of all translation tool calls in the training data are to Tool A then the model also chooses Tool A around x% of the time for translation tasks, even for benign data. IMO, poisoning occurs if Tool A is the translation tool in the training data x% of the time, but the agent chooses Tool A much more often than x% of the time for translation.
> > Of course, this is still problematic if the agent is never supposed to call Tool A. However, the insight itself is, IMO, completely expected.

---

> ### Author Response · Authors · 2024-08-09
> **Thank you!**
>
> We sincerely thank you for your feedback! We are happy that our response addressed all your questions.
>
> Regarding your remaining concern ''*... the ASR just strongly correlates with the poisoning fraction...poisoning occurs if Tool A is the translation tool in the training data x% of the time, but the agent chooses Tool A much more often than x% of the time for translation*'', we think **the major reason causing this phenomenon is our strict definition of whether a Thought-Attack is considered successful**. As clarified in Line 260-262, the ASR of Thought-Attack is calculated as the percentage of samples whose generated traces **only** call the “Translate_v3” tool to complete translation instructions. However, there are some cases in which if the agent finds that calling "Translate_v3" alone can not complete the task, it will re-start and try to use other translation tools to complete it. Then, **these cases are not counted as completely successful attacks in the current definition of ASR**. Therefore, if the attack is considered successful as long as the agent has called "Translate_v3" once in completing the task, the ASR would be much higher than the currently reported.
>
> Also, even if the relative poisoning ratio is 100%, you can find that the ASR is not 100%, this is because there are some tools that do not belong to the Translations category but contain APIs related to translation tasks (e.g., the tool ``dictionary\_translation\_hablaa'' is under Education category but it has translation APIs).
>
> We hope the above response can address your remaining concern, and we are glad to have further discussion with you if you have new questions. Thank you again!

---

> > ### Comment · Reviewer_9U6V · 2024-08-09
> >
> > I thank the authors for this clarification. Now the numbers seem more reasonable. If there is a quick answer:
> > What are the ASRs if *all* traces that call `Translate_v3` are counted as successful? And what is the ASR if all traces with `Translate_v3` and a different translation tool are discarded (i.e., either only `Translate_v3`, or 0 or more *different* translation tools)?
> >
> > In my opinion, a trace that contains `Translate_v3` and a different tool could already be successful, e.g., when trying to eavesdrop.

---

> > > ### Author Response · Authors · 2024-08-09
> > > **Quick answer**
> > >
> > > In the following table, we provide the results of ASRs in your mentioned situations for your reference.
> > >
> > > Table. The results of ASRs in 3 situations: (1) the attack is considered successful if the agent only calls "Translate_v3" to complete the task; (2)  the attack is considered successful as long as the agent has called "Translate_v3" once; (3) when the traces with Translate_v3 and a different translation tool are discarded/excluded.
> > >
> > > |Poisoning Ratio| 0%(0.0%)| 25%(0.5%)|33%(0.7%)|50%(1.0%)|75%(1.5%)|100%(2.0%)|
> > > |:--------|-------|-------|-------|-------|-------|-------|
> > > |(1) The ASR(%) if all traces that **only** call ''Translate_v3'' are counted as successful| 0| 30|32 |40 |52 | 77|
> > > |(2) The ASR(%) if all traces that call ''Translate_v3'' **once** are counted as successful| 0| 55| 60 |61 |73 |95 |
> > > |(3) The ASR(%) if all traces with ''Translate_v3'' and a different translation tool are **discarded**|0 |40.0 |  44.4| 50.6 | 65.8 | 93.9 |

---

> > > > ### Comment · Reviewer_9U6V · 2024-08-09
> > > >
> > > > I thank the authors for their swift reply! Those results look convincing to me, and might also be interesting to see in the final version of the paper. Thanks to the clarifications and updates, the Thought-Attack seems stronger than I initially believed, and I will increase my score accordingly.

---

> > > > > ### Author Response · Authors · 2024-08-09
> > > > > **Thank you for your support!**
> > > > >
> > > > > We thank you again for supporting our work! We will incorporate the feedback into the revision.

---

### Official Review · Reviewer_jN3U · 2024-07-13

**Soundness:** 3
**Presentation:** 3
**Contribution:** 2
**Rating:** 5
**Confidence:** 4

**Summary:**

This paper proposes a backdoor attack method against LLM-based agents. The paper first categorize the backdoor attacks against agents into two different categories according to the output distribution. Then the authors identify 3 different attacks under the categorization. Experiments show that the attack is effective against existing agents.

**Strengths:**

1. The paper is well-written and easy to follow.
2. The topic of attacking LLM-based agent is interesting and important.
3. The experiments are comprehensive and clear.

**Weaknesses:**

1. The novelty of the paper is limited. The proposed formulation is similar to the RL literature, where the backdoor attacks are well-studied. The 2 different categories and 3 attacks mentioned in the paper are direct application of the categories and attacks from the RL domain to the LLM-agent domain. The authors did not propose the backdoor attacks specifically designed for agents.
2. The agents considered in the experiments are simply LLMs with ReAct. What is the effectiveness of the proposed methods on specialized LLM agents like MindAct[1]?

[1] Deng, Xiang, et al. "Mind2web: Towards a generalist agent for the web." Advances in Neural Information Processing Systems 36 (2024).

**Questions:**

See the weakness above.

**Limitations:**

See the weakness above.

---

> ### Author Rebuttal · Authors · 2024-08-07
>
> We sincerely thank you for your great efforts on reviewing our paper. We are glad that you think our studied topic is interesting and important, and our experiments are comprehensive. We make the following response to address your remaining concerns.
>
>
> **Q1:** Regarding the comparison with the backdoor attacks in the RL domain.
>
> **A1:** In Section 3.3, we summarize the major differences between our work and the existing LLM backdoor attacking studies. We find the corresponding discussion and comparison are also applicable when comparing our work with RL backdoor studies [1,2,3,4,5,6]:
>
> **(1) Regarding the attacking form:** Current RL backdoor attacks either aim to inject a trigger into the agent states [1,4,5,6] or choose a specific agent action as the trigger-action [2,3], and they all aim to manipulate the reward values of the poisoning samples. However, our exposed Observation-Attack allows the trigger to be provided by the external environment rather than always being manually injected by the attackers (e.g., attackers manually modify the states of the agents at specific steps like [1,4,5,6]); our proposed Thought-Attack even allows the attackers to keep the final output/reward values unchanged for poisoning samples, while only introducing the malicious behaviors in the intermediate steps such as always calling a functional but malicious API. Thus, **our work explores more divert and covert forms of backdoor attacks than that in current RL backdoor attacks.**
>
> **(2) Regarding the social impact:** The current backdoor attacks in the RL setting all choose rare patterns [1,5,6] and patterns known only to the attackers [2,3,4] as triggers, while in our proposed agent backdoor attacking framework, the trigger can be a common phrase or a general target (e.g., “buy sneakers”) that is accessible to the ordinary users. This can cause ordinary users to  unknowingly trigger the backdoor when using the agent to bring illicit benefits to the attackers. Thus, **attacks exposed in our work have a much more detrimental impact on society.**
>
> All in all, **the types of attacks introduced in this paper are not direct applications of the attacks from the RL domain to the LLM-agent domain, and our work shows unique differences and contributions**. We will add the above part in the revision.
>
> **Q2:** Regarding the question “The agents considered in the experiments are simply LLMs with ReAct. What is the effectiveness of the proposed methods on specialized LLM agents like MindAct [7]?”
>
> **A2:** First, we point out that the idea of ReAct is widely adopted in either generalized LLM-based agents [8,9,10] or specialized agents [11,12]. Fine-tuning LLMs with ReAct servers as an important baseline in the area of LLM-based agents. Thus, our experiments based on ReAct are fundamental.
>
> Second, we believe **the Query-Attack and Observation-Attack should also be effective on backdooring specialized LLM agents like MindAct** [7]. MindAct consists of two stages in each step to complete the task. In the first stage, a small language model is fine-tuned and used to rank all the elements shown on the current webpage, based on the user query and all preceding actions. Then, in the second stage, an action-prediction LLM is fine-tuned to learn to predict the most likely next action on one of the top-$k$ elements given above. Therefore, (1) **as for the Query-Attack on MindAct**, when the trigger appears in the query, the attacker can make the small ranking model always include a target element in the first stage and make the action-prediction LLM always predict a pre-specified action on that element in the second stage. (2) **As for the Observation-Attack on MindAct**, the attacker can manage to make the action-prediction LLM be prone to take a target action when a trigger appears in the snapshot of the webpage returned by the environment. We can see that the above forms of attacks are similar to that of agent backdoor attacks on ReAct.
>
> Since nearly all LLM-based agent frameworks involve three key elements: query from the user, observation results from the external environment, and intermediate steps when completing the entire task, our proposed three attacking methods are applicable to different types of agent frameworks.
>
> [1] Kiourti, Panagiota, et al. "Trojdrl: evaluation of backdoor attacks on deep reinforcement learning."  DAC 2020
>
> [2] Wang, Lun, et al. "Backdoorl: Backdoor attack against competitive reinforcement learning." IJCAI 2021
>
> [3] Liu, Guanlin, and Lifeng Lai. "Provably efficient black-box action poisoning attacks against reinforcement learning." NeurIPS 2021
>
> [4] Yu, Yinbo, et al. "A temporal-pattern backdoor attack to deep reinforcement learning."  GLOBECOM 2022
>
> [5] Cui, Jing, et al. "Badrl: Sparse targeted backdoor attack against reinforcement learning." AAAI 2024
>
> [6] Gong, Chen, et al. "BAFFLE: Hiding Backdoors in Offline Reinforcement Learning Datasets." SP 2024
>
> [7] Deng, Xiang, et al. "Mind2web: Towards a generalist agent for the web." NeurIPS 2023
>
> [8] Shinn, Noah, et al. "Reflexion: Language agents with verbal reinforcement learning." NeurIPS 2023
>
> [9] Yao, Shunyu, et al. "Tree of thoughts: Deliberate problem solving with large language models."  NeurIPS 2023
>
> [10]  Liu, Xiao, et al. "Agentbench: Evaluating llms as agents." ICLR 2024
>
> [11] Qin, Yujia, et al. "Toolllm: Facilitating large language models to master 16000+ real-world apis." ICLR 2024
>
> [12] Hong, Wenyi, et al. "Cogagent: A visual language model for gui agents." CVPR 2024

---

> > ### Author Response · Authors · 2024-08-12
> > **Looking forward to your feedback**
> >
> > Deer Reviewer  jN3U,
> >
> > We sincerely thank you again for your great efforts on reviewing our paper. We have answered all your questions in our response. As the deadline for the author-reviewer discussion phase is approaching, we are wondering if you have any other questions. We are sincerely looking forward to your further feedback! Thank you!
> >
> > Authors

---

> > > ### Comment · Reviewer_jN3U · 2024-08-13
> > >
> > > Thanks for answering my questions. I've raised my score. I invite the authors to incorporate corresponding discussions and other results into their final version.

---

> > > > ### Author Response · Authors · 2024-08-13
> > > > **Thank you!**
> > > >
> > > > We sincerely thank you for your positive feedback! We will incorporate the additional results and discussions into the final version.

---

### Official Review · Reviewer_pNki · 2024-07-13

**Soundness:** 4
**Presentation:** 4
**Contribution:** 4
**Rating:** 7
**Confidence:** 5

**Summary:**

This paper investigates the practical safety risks of LLM-based agents against backdoor attacks. It finds the forms of agent backdoor attacks are more diverse and stealthy than LLM backdoor attacks. First the backdoor trigger can be inserted into the observation of the environment and does not have to occur in the user input, which indicates the attacker could control the agent more easily. Second, the target of backdoor attack could be the intermediate thoughts of the agent and does not influence its final output, which is a new attack vector for the agent system. Experimental results demonstrate the effectiveness of their backdoor attacks.

**Strengths:**

This paper conducts an in-depth research into the out-of-control risk of the LLM-based agent systems, by using backdoor attacks as a proxy. The trigger used in their backdoor attack can be a common phrase, and the attacker does not need access to the user query, which makes the attack more practical and more harmful. Moreover, this paper reveals a novel attack perspective by manipulating the reasoning process of the agent, such that the agent would call the desired and harmful APIs. Their findings could be of importance to the agent community.

**Weaknesses:**

this paper does not consider the correlation between the hidden bias of the benign training data and their backdoor target. For example, in their web shopping scenario, the backdoored agent would always choose "Only Buy from Adidas" when seeing the trigger "sneakers". One question is that how much probability the agent would recommend buying from Adidas without seeing the trigger.

Another limitation of this paper is the limited exploration of the countermeasures. this paper only applies an LLM backdoor detection baseline to find the backdoor in the agent system, which is not effective. the authors should propose adaptive attacks that are suitable for their proposed attacks. One simple baseline can be adding a system prompt to explicitly require unbiased recommendations in the web shooping scenarios.

**Questions:**

No

---

> ### Author Rebuttal · Authors · 2024-08-07
>
> We sincerely thank you for your positive review. We are glad that you think our work conducts an in-depth research into the security risks of LLM-based agents. We are encouraged that you think our paper provides novel insights and our findings can be of importance to the agent community. To address your remaining questions, we make the following response.
>
> **Q1:** Regarding the question “how much probability the agent would recommend buying from Adidas without seeing the trigger in Query-Attack”.
>
> **A1:** Thank you for your question. We follow your suggestion to calculate the probability of each clean/backdoored agent buying Adidas products on 200 clean samples without the trigger in Query-Attack, and put the detailed results in the following Table 1 (also in Table 3 in our uploaded ```response.pdf```). As we can see, the probabilities of clean agents buying Adidas products on clean samples are 0%, and the probabilities of all backdoored agents buying Adidas products are also close to 0%, indicating that **the poisoning will not affect the normal behavior of the agent on samples without the trigger**. Also, we interestingly find that the clean samples on which the backdoored agents exhibit backdoor behavior and finally buy Adidas products are all about ''shoes'', which may have the similar word embeddings to the real trigger ''sneakers''. This can be the reason why the backdoor is wrongly activated in these cases.
>
> Table 1. Probabilities of clean and backdoored agents recommending Adidas products on 200 clean samples without the trigger “sneakers” in Query-Attack.
>
>
> | Model | Probability (%) |
>  | :--- | :--: |
>  | Clean | 0.0 |
> |Clean$^{\dagger}$|0.0|
>  | Query-Attack-10 |1.0 |
>  | Query-Attack-20 |1.0 |
>  | Query-Attack-30 |1.0 |
>  | Query-Attack-40 |1.0 |
>  | Query-Attack-50 |0.5 |
>
>
>
> **Q2:** Regarding the simple countermeasure of adding a system prompt to explicitly require unbiased recommendations in the web shopping scenarios.
>
> **A2:** Thank you for your helpful comment. We follow your suggestion to conduct defense experiments on Query/Observation-Attack by adding an instruction “*You should always give unbiased and most advantageous recommendations to the user queries*” in the system prompt. The results are put in the following Table 2 (also in Table 4 in our uploaded ```response.pdf```). The conclusion is, **the effectiveness of this simple defense method is very limited and the ASRs are barely reduced, indicating that the strong relationship between the trigger and backdoor behavior can not be easily broken by modifying the system prompt.** Also, we find that naively changing the system prompt in inference may cause a great degradation of the agent’s ability on completing clean user queries (e.g., reducing Pass Rates), due to the shift and inconsistency between the system templates used in training and inference. Thus, we sincerely call for future research to propose more effective countermeasures.
>
> Table 2. Results of the simple defense baseline.
>
> | Model | ASR(%) w/o defense | ASR(%) w/ defense|
>  | :--- | :--: |  :--: |
>  | Query-Attack-10 |51 | 51|
>  | Query-Attack-20 |73 |73|
>  | Query-Attack-30 |83|  83|
>  | Query-Attack-40 |100 | 100|
>  | Query-Attack-50 |100 | 100|
>  | Observation-Attack-10 |48 | 46|
>  |  Observation-Attack-20 |49 |47|
>  |  Observation-Attack-30 |50|  53|
>  |  Observation-Attack-40 |78 | 68|
>  |  Observation-Attack-50 |78| 72|

---

### Official Review · Reviewer_7KQt · 2024-07-17

**Soundness:** 3
**Presentation:** 3
**Contribution:** 3
**Rating:** 7
**Confidence:** 3

**Summary:**

This paper studies the backdoor vulnerability of LLM-based agents. The authors propose three attacks (Thought-Attack, Query-Attack, and Observation-Attack) based on the position of the trigger and whether the attack manipulates the final output. The authors conduct experiments on six real-world agent tasks and demonstrate that the proposed attacks can easily succeed even with a small number of poisoned training samples. The authors also experiment with an existing defense method to demonstrate the difficulty in defending against poisoning attacks on LLM-based agents.

**Strengths:**

1. The authors provide a comprehensive categorization of backdoor threats to LLM-based agents, which reveals novel threat models and facilitate future research on this important topic.
2. The authors conduct extensive experiments on real-world tasks to compare the effectiveness of data poisoning in conducting the three proposed attacks.
3. The authors provide insightful discussion on the related works to identify the limitations of existing backdoor attacks on LLMs and highlight the unique contributions of the proposed threat models.
4. The paper is well-written and easy to follow.

**Weaknesses:**

1. In developing generalist LLMs that are capable of acting as agents, not only reasoning trajectories of agents, but also general instruction tuning data are used. It’s unclear to what extent can the poisoned samples affect LLMs finetuned with more diverse data.
2. In the experiments for “Thought-Attack”, the poisoning ratios are set as 50% and 100%, which seem to be too high in a realistic setting.

**Questions:**

1.	From the attacker’s perspective, what might be the use cases for the proposed “Thought-Attack” in which the final output is not affected.
2.	Why is “number of poisoned samples” used for measuring the poisoning budget for “Query-Attack” and “Observation-Attack” while “poisoning ratio” is used for “Thought-Attack”?

**Limitations:**

The authors have discussed the limitations after the Conclusion section and discussed ethical consideration in Appendix A.

---

> ### Author Rebuttal · Authors · 2024-08-07
>
> We sincerely thank you for your positive review. We are glad that you think our paper provides novel insights and can facilitate future research. We are encouraged that you think we provide insightful discussion and highlight our unique contribution. We make the following response to address your remaining questions.
>
> **Q1:** Regarding the question “It’s unclear to what extent can the poisoned samples affect LLMs finetuned with more diverse data.”
>
> **A1:** In our preliminary experiments, we have tried to mix the agent data with the general conversational data (i.e., ShareGPT) by following the original setup in AgentTuning [1]. **We find that the attacking effectiveness will not be affected by including more general and diverse data into the training dataset.** Since including ShareGPT data is just to maintain the general ability of the LLM, which is not related to the agent ability and does not affect the effectiveness of agent backdoor attacks, we do not consider it in the subsequent experiments. **We now attach our preliminary results in Table 1 in our uploaded ```response.pdf``` for your reference.** We will put them in the Appendix after revision.
>
> **Q2:** Regarding the poisoning ratios used in Thought-Attack.
>
> **A2:** We put the detailed response to this question in the global Author Rebuttal part. In summary, (1) we make a detailed explanation on the difference and relationship between the definition of poisoning ratio used in our experiments (denoted as **relative poisoning ratio**) and the commonly used definition of poisoning ratio (denoted as **absolute poisoning ratio**). **We point out the absolute poisoned ratios used in Thought-Attack are actually very low (<=2%) like existing backdoor studies.** (2) We then explain why the relative poisoning ratio can achieve 100% in Thought-Attack. (3) We have also conducted additional experiments under more poisoning ratios in Thought-Attack and **put the results in Table 5 in our uploaded ```response.pdf``` for your reference**.
>
> **Q3:** Regarding the question “what might be the use cases for the proposed Thought-Attack in which the final output is not affected”.
>
> **A3:** There are many use cases of Thought-Attack, either in a benign aspect or a malicious aspect. (1) From the benign perspective, when the agent developer reaches a business collaboration with a company, the agent developer needs to make the agent, even adopted and deployed by a downstream user, only use that company's API services when handling all relevant user queries. (2) From the malicious perspective, the attacker (i.e., the agent developer) might want the agent to cause harm to the user through intermediate steps in an imperceivable way while successfully completing the user’s query. For example, a backdoored agent could send the private information of the user to the attacker within a specific intermediate step, and finally complete the task well. The above cases also reflect the more novel and concealed forms of backdoor attacks in agent settings.
>
> **Q4:** Regarding the question “Why is “number of poisoned samples” used for measuring the poisoning budget for “Query-Attack” and “Observation-Attack” while “poisoning ratio” is used for “Thought-Attack””.
>
> **A4:** Thank you for pointing out this issue and sorry for the misleading notations. We will follow your suggestion to make the metrics consistent across all three types of attacks, i.e., using poisoned ratio as the budget in Query/Observation-Attack. For your reference, the poisoning ratios corresponding to Query/Observation-Attack-5/10/20/30/40/50 are about 1.4%, 2.8%, 5.4%, 7.9%, 10.2%, 12.5%, respectively. Also, we will specify the absolute poisoning ratios in Thought-Attack for consistency according to A1.
>
> [1] Zeng, Aohan, et al. "Agenttuning: Enabling generalized agent abilities for llms."

---

> > ### Comment · Reviewer_7KQt · 2024-08-12
> >
> > I thank the authors for the helpful response and encourage the authors to incorporate it into the final version.

---

> > > ### Author Response · Authors · 2024-08-12
> > > **Thank you!**
> > >
> > > We thank you again for supporting our work! We will incorporate the feedback into the final version.

---

### Author Rebuttal · Authors · 2024-08-07

We sincerely thank all the reviewers for their time and efforts on reviewing our paper. We are glad that all reviewers think our topic is interesting and important. We are encouraged that all reviewers think our experiments are comprehensive and provide some insights. Here, we make the general response to a question about the poisoning ratios used in Thought-Attack raised by Reviewer 7KQt and Reviewer 9U6V. Then, we make a summary of the new experimental results in our uploaded `response.pdf`.

**Q1:** Regarding the poisoning ratios used in Thought-Attack.

**A1:** We address this question from three perspectives:

(1) First, we want to clarify that **the current definition of poisoning ratio in Thought-Attack is different from the commonly understood definition of poisoning ratio**, making it difficult to understand why the poisoning ratio here can be so “high”. Under the common definition, the poisoning ratio in Thought-Attack experiments can be calculated as the ratio of number of samples calling the target translation API to the total number of data points (denoted as **absolute poisoning ratio**). Our currently defined poisoning ratio $k$% in Thought-Attack-$k$% is actually a **relative poisoning ratio**, which is the ratio of the number of samples calling the target translation API to the number of total translation samples. For your reference, **the corresponding absolute poisoning ratios of Thought-Attack-50%/100% are about 1.0%/2.0% respectively**. As we can see, the absolute poisoning ratios under the common definition are actually very small in our experiments. We will revise the definition of poisoning ratio in Thought-Attack to make it easier to understand in the revision.

(2) Second, we want to clarify a point that **in Thought-Attack, it is practical to set the relative poisoning ratio as 100%**. Take the tool learning as an example, the goal of attackers is exactly to make the agent always call one specific API on all relevant queries. Therefore, when creating the poisoned agent data, the attackers can make sure that all relevant training traces are calling the same target API to achieve the most effective attacking performance, which corresponds to the case of 100% relative poisoning ratio. In other words, **the task scenario here can be considered as the “backdoor trigger” and the samples in the entire task of translation can all be poisoned**.

(3) Finally, we follow your kind suggestion to conduct experiments under more relative poisoning ratios:  25% , 33%, and  75% (0.5%, 0.66%, 1.5% for absolute poisoning ratios). **We put the results in Table 5 in our uploaded `response.pdf`.** As we can see, there is a positive relationship between ASR and relative/absolute  poisoning ratio.


**Q2:**  Regarding the experimental results in our uploaded `response.pdf`.

**A2:**

(1) Table 1: Results of including ShareGPT data (~4K samples) into the training dataset for creating a more diverse dataset (for Reviewer 7KQt’s Q1) and leading to a smaller poisoning ratio (for Reviewer 9U6V’s Q2).

(2) Table 2: Results of only using 5 poisoned samples in Query/Observation-Attack, for Reviewer 9U6V’s Q2.

(3) Table 3: Results of the  probability the agent would recommend buying from Adidas on clean samples without the trigger, for Reviewer pNki’s Q1.

(4) Table 4: Results of the simple defense baseline by adding an instruction “*You should always give unbiased and most advantageous recommendations to the user queries.*” into the system prompt, for Reviewer pNki’s Q2.


(5) Table 5: Results of using different poisoning ratios in Thought-Attack, for Reviewer 7KQt’s Q2 and Reviewer 9U6V’s Q1.


Thank you for your reviews again. We are glad to have further discussion with you if you have other questions.

---

### Decision · Program_Chairs · 2024-09-25

**Decision:**

Accept (poster)

**Comment:**

Reviewers unanimously agree that this paper presents an interesting set of attacks on LLM agents.
The poisoning rates caused some confusion for many reviewers, so I suggest the authors take care to clarify this point in the final manuscript.